# A New BODIPY-Based Receptor for the Fluorescent Sensing of Catecholamines

**DOI:** 10.3390/molecules29153714

**Published:** 2024-08-05

**Authors:** Roberta Puglisi, Alessia Cavallaro, Andrea Pappalardo, Manuel Petroselli, Rossella Santonocito, Giuseppe Trusso Sfrazzetto

**Affiliations:** 1Department of Chemical Sciences, University of Catania, Viale Andrea Doria 6, 95125 Catania, Italy; roberta.puglisi@unict.it (R.P.); alessia.cavallaro7@studium.unict.it (A.C.); andrea.pappalardo@unict.it (A.P.); 2Research Unit of Catania, National Interuniversity Consortium for Materials Science and Technology (I.N.S.T.M.), Viale Andrea Doria 6, 95125 Catania, Italy; 3Institute of Chemical Research of Catalonia (ICIQ), Av. Països Catalans 16, 43007 Tarragona, Spain; mpetroselli@iciq.es

**Keywords:** catecholamine neurotransmitters, BODIPY, healthcare, DFT calculations, optical fiber

## Abstract

The human body synthesizes catecholamine neurotransmitters, such as dopamine and noradrenaline. Monitoring the levels of these molecules is crucial for the prevention of important diseases, such as Alzheimer’s, schizophrenia, Parkinson’s, Huntington’s, attention-deficit hyperactivity disorder, and paragangliomas. Here, we have synthesized, characterized, and functionalized the BODIPY core with picolylamine (BDPy-pico) in order to create a sensor capable of detecting these biomarkers. The sensing properties of the BDPy-pico probe in solution were studied using fluorescence titrations and supported by DFT studies. Catecholamine sensing was also performed in the solid state by a simple strip test, using an optical fiber as the detector of emissions. In addition, the selectivity and recovery of the sensor were assessed, suggesting the possibility of using this receptor to detect dopamine and norepinephrine in human saliva.

## 1. Introduction

Neurotransmitters work as crucial endogenous primary chemical messengers, transporting information between biological cells in the mammalian central nervous system and indicating states of health and disease [1].

An important catecholamine neurotransmitter (CA-NTs) is dopamine (DA), a derivative of phenethylamine produced by the adrenal medulla, which plays a vital role in numerous brain functions. Changes in DA levels are linked to various conditions, such as schizophrenia; Alzheimer’s, Parkinson’s, and Huntington’s disease; attention-deficit hyperactivity disorder (ADHD); and paragangliomas [2,3,4,5,6]. Therefore, monitoring DA levels is essential for assessing human health. Typically, DA concentrations are around 20 ng/mL in plasma, 18.9 pg/mL in saliva, and between 0.2 and 1 mg/mL in urine [7].

Another important CA-NT is norepinephrine (NE), released from nerve endings in the sympathetic nervous system and certain areas of the cerebral cortex [8,9]. Also known as noradrenaline, it is an endogenous hormone secreted by the adrenal medulla. NE plays a critical role in regulating arousal, attention, mood, learning, memory, and stress response [10]. Peripherally, NE increases heart rate, cardiac contractility, vascular tone, renin–angiotensin system activity, and renal sodium reabsorption. The NE concentration is around 30 pg/mL in the saliva [11]. Therefore, measuring the NE levels in biological fluids is crucial for understanding its physiological functions and improving disease diagnosis.

Clinical diagnostics plays a crucial role in patient healthcare by providing vital information about the extent, causes, and consequences of diseases through clinical test data. Most of these diagnostics involve blood samples, an invasive procedure that can cause discomfort and requires skilled personnel for collection and handling. To address these issues, saliva offers a non-invasive, easy, and convenient alternative for clinical analyses. This method reduces patient stress, allows rapid sample collection, and lowers the risk of contamination. Saliva-based testing has proven valuable for diagnosing diseases by detecting antigens and biomarkers, facilitating frequent and non-invasive testing [12]. Saliva contains a diverse range of biological molecules from the salivary glands, external substances, and microorganisms, which serve as diagnostic targets for various diseases. The concentrations of clinically relevant biomarkers in saliva can reflect the patient’s health status.

Various analytical methods have been utilized to determine norepinephrine (NE) and dopamine (DA) [13,14,15,16]. Due to NE’s electroactive nature, electrochemical methods, mainly based on polymer films (see Ref. [17]), have garnered significant attention and are more frequently applied than other techniques. However, the similar electrochemical behavior of dopamine (DA) and epinephrine (EP), as well as interference from substances like ascorbic acid, poses challenges to these methods [18]. Chromatographic techniques, such as capillary electrophoresis and high-performance liquid chromatography, have also been used for NE determination, but they are often tedious, time-consuming, labor-intensive, and expensive, involving complex procedures. Molecular sensors bypass these problems due to their fast response and relatively easy detection methods [19,20]. In particular, optical nanocomposite sensors based on polyurethane foam and gold nanorods for the solid-phase spectroscopic determination of catecholamines [21] and carbon quantum dot (QD) arrays of CdTe coated with thioglycolic acid (TGA) have been used for the identification and discrimination of catecholamines [22,23]. Recently, an optical array based on gold nanoparticles able to detect different catecholamines in mixtures, as well as in urine samples, with micromolar limits of detection, has been reported [24]. In 2019, Zhang and co-workers reported a fluorescent sensor based on a quinolone fluorophore with a boronic acid recognition element, which showed high affinity for catecholamines and, in particular, a turn-on response to norepinephrine. This fluorescent sensor for norepinephrine showed the highest affinity for live-cell assays. To the best of our knowledge, no examples of molecular sensors able to distinguish DA and NE simultaneously in saliva have been reported [25].

In this context, a new fluorescent BODIPY (BDPy) receptor bearing a picolylamine arm, able to interact by non-covalent interactions with DA and NE, is reported here. Sensing studies were performed in solution using fluorescence titrations and supported by DFT calculations. A strip test was fabricated to also detect these catecholamines in the solid state, valuating its selectivity with respect to the other analytes contained in human saliva.

## 2. Results and Discussion

BODIPY probes were synthetized following the reaction pathway shown in Figure 1 (see Materials and Methods for details).

In particular, starting with kryptopyrrole, in the presence of chloroacetyl chloride and after the addition of triethylamine and boron trifluoride, the corresponding fluorophore (**1**) was obtained. BDPy-pico (**2**) was synthesized starting with (**1**) through reaction with excess 2-picolylammine in the presence of KI and K_2_CO_3_.

^1^H, ^13^C NMR, and ESI-MS analyses confirmed the chemical structure of (**2**). Optical characterization was performed in chloroform solution. Figure 1 shows the absorption and emission spectra of 1 μM solution. Table 1 shows the absorption (Abs) and emission (Em) values of BDPy (**1**) and (**2**) and the relative quantum yields and Stokes shifts (nm) [26]. In particular, the UV-Vis spectrum shows an intense band centered at 530 nm (ε 61,668 M^−1^·cm^−1^) with a shoulder at 500 nm. The emission spectrum was obtained with excitation at 500 nm. A strong emission at 550 nm can be detected, with a Stokes shift of 50 nm, ideal for sensing applications.

Sensing studies were performed in chloroform solution. Unfortunately, ^1^H NMR titrations involving (**2**) and DA or NE did not yield significant information about the functional groups involved in the complexation, probably caused by the high concentrations required for NMR experiments. The ability of (**2**) to detect DA and NE was assessed by fluorescence titrations in a concentration range of 1 × 10^−7^ M to 1 × 10^−4^ M for DA and NE, using 1 × 10^−6^ M of the receptor (the UV-Vis spectra of (**2**) in the presence of DA and NE do not show significant changes).

Figure 2a,b show the fluorescence titrations of (**2**) vs. DA and NE, respectively (see the Appendix A). The binding constants and detection limits were determined using fluorescence titrations by observing the emission changes at 550 nm as DA or NE was progressively added (Figure 2). In particular, (**2**) shows a higher affinity in solution for DA than NE, with a difference in the binding constant values at an order of magnitude of ca. 2.5 (logK 6.19 and 3.79 for DA and NE, respectively). The LODs obtained for the titrations in solutions are very promising (36 nM for DA and 27 nM for NE); considering the physiological levels of DA and NE, this study is very promising for selective detection of these CA-NTs in biological fluids.

To shed light on the nature of the host–guest (HG) complexes between (**2**) and the involved guests, DA and NE, we performed a DFT study to investigate the complex geometries and non-covalent interactions involved in the complex formation [28,29]. A theoretical study was performed to (i) overcome the issues found during the NMR titration and (ii) access useful structural information on the HG complexes in order to shed light on the high selectivity of (**2**) towards DA, although it is structurally similar to NE. The presence of several hydrogen bond donors and acceptors in (**2**) and the guests (DA and NE) forced us to perform an initial conformational screening of the HG complexes (see Appendix A). The most stable conformer for the **DA@2** complex shows non-covalent interactions between DA’s catechol moiety and the boron difluoride group of (**2**), while DA’s terminal amino group interacts with the secondary amine of the host (Appendix A—Conformation 2). This conformation is predominant, with a Boltzmann distribution higher than 99% due to the ΔE value of −2.9 kcal/mol with respect to the second main conformer (Appendix A—Conformation 1). Similarly, the **NE@2** complex shows the same main conformation but with a Boltzmann distribution of 82% due to the lower ΔE value (−0.90 kcal/mol) compared to the second main conformer (Appendix A). Although the catechol moiety of (**2**) binds the boron difluoride group of the host, as previously described for **DA@2**, the terminal amino group in NE is now not involved in any hydrogen bond, unlike in the **DA@2** complex. Instead, it is the terminal hydroxyl group in NE that interacts with the secondary amine and the pyridine moiety of (**2**) (Appendix A). NE’s second main conformer (with a Boltzmann distribution of 18%) instead shows reverse binding compared to the previous type. Indeed, the guest’s catechol group now interacts with the secondary amine and pyridine moieties, while the terminal amino and hydroxyl groups synergically bind the boron difluoride moiety of (**2**) (see Appendix A).

The complexation energy (E_complex_) was calculated for the main conformers of the **DA@2** and **NE@2** complexes, which showed Boltzmann distributions of 99% and 82%, respectively, surprisingly unveiling a huge difference (ΔE = 15.4 kcal/mol). A stabilization energy of 36.7 kcal/mol was found for the **DA@2** complex, while a value of 21.3 kcal/mol was calculated for the **NE@2** complex. A similar complexation energy (20.4 kcal/mol) was found for the less abundant conformer in the **NE@2** complex (with a Boltzmann distribution of 18%), highlighting once again the much higher affinity of **2** for DA (Appendix A). These findings are in agreement with the binding constants experimentally measured through fluorescent titration (Figure 2), where the affinity of (**2**) for DA was found to be almost three orders of magnitude higher than its affinity for NE.

The higher number of hydrogen bonds in **NE@2** compared to those in the **DA@2** complex (4 vs. 3, respectively—Figure 3) makes the elucidation of the higher affinity calculated for **DA@2** (ΔE = 15.4 kcal/mol) counterintuitive and non-obvious. It is well known that hydrogen bonds are directional bonds and are strongly affected by the orientation of the systems involved [30]. In light of this, a series of structural parameters, such as the length and angle, of the hydrogen bonds involved in both HG complexes were theoretically measured (Table 2). The hydrogen bonds between the catechol moiety and the boron difluoride group (Figure 3—hydrogen bonds A and B) are almost identical in terms of length (Δd < 0.03 Å) in both HG complexes. Instead, differences in terms of spatial orientation were observed for hydrogen bond A, which shows a more directional angle in **DA@2** than the **NE@2** complex. This is quite evident in observing the angle related to the BF⋯H interaction (hydrogen bond A), which is 137.03° for **DA@2**, while a wider angle (145.57°) was found in **NE@2** (Table 2—Entries 1 and 2), suggesting a weaker interaction. A similar outcome was also found for hydrogen bonds C-E, but the different hydrogen bond networks in **DA@2** and **NE@2** do not allow a direct comparison, as previously made for hydrogen bonds A and B. However, the double hydrogen bond in the **NE@2** complex with the secondary amine and pyridine (Figure 3—hydrogen bonds D and E) appears relatively distorted, as confirmed by the angles of 150.92° and 157.60° found for hydrogen bonds D and E, respectively (Table 2—Entries 4 and 5). A less distorted hydrogen bond with the same secondary amine is instead observed in the **DA@2** complex (Figure 3—hydrogen bond C), as highlighted by the angle of 100.97° found for hydrogen bond C (Table 2—Entry 3). To further shed light on the reasons for the higher affinity observed for the **DA@2** complex, we performed a Mulliken population analysis study on each HG complex. No huge differences were observed in the hydrogen bond between catechol and boron difluoride groups (Appendix A—Entry 9–12), highlighting a comparable electrostatic contribution in both complexes of hydrogen bonds A and B. Instead, a divergent electrostatic contribution was observed in the formation of hydrogen bonds C-E, but once again, the different hydrogen bond networks in **DA@2** and **NE@2** (Figure 3) made direct comparison quite challenging. Some comparisons and information can be obtained by taking into account the NH group of the secondary amine moiety in (**2**), which is directly involved in a single hydrogen bond in both complexes (Figure 3—hydrogen bonds C and D). The H atom in the NH group shows a Mulliken partial charge of 0.314 in **DA@2**, while a less positive value (0.279) is found in **NE@2** (Appendix A—Entry 3), confirming a stronger hydrogen bond with DA’s terminal amino group compared to that with the terminal hydroxyl group in NE, probably due to the better spatial orientation. This outcome is further confirmed by the Mulliken partial charge of the N atom, which shows a value of −0.562 in **DA@2**, while a more positive value (−0.528) was found for **NE@2** (Appendix A—Entry 2), once again highlighting the formation of stronger hydrogen bonds in the **DA@2** complex compared to the ones formed in the **NE@2** complex.

### Sensing by the Strip Test

In order to obtain a practical device for monitoring DA and NE levels, a strip test was performed. In brief, 2 μL of BDPy (**2**) solution (1 mM in CHCl_3_) was dropped onto polyamide filter paper, and then the solvent was evaporated. Next, the solution under study was transferred using a common swab, and an optical fiber was employed as the detector, with excitation at 365 nm (Figure 4). In particular, two different water solutions of DA and NE (ranging from 10 μM to 1 nM) were deposited on the BDPy probe (**2**) with a polycellulose cartridge. Experiments with the strip test were all conducted at a pH = 7 (the average pH of human saliva).

Figure 5 shows the response of the BDPy (**2**) probe to exposure to DA (blue) and NE (orange) and demonstrates that the BDPy probe is able to distinguish between DA and NE in a concentration range between 1 μM and 1 nM, while higher concentrations (10 μM) are not distinguished between. This result is very promising for the purpose of sensing these catecholamines in human saliva. We noted that the emission change in (**2**) in the presence of catecholamines is different from that in the strip test. This behavior has been widely demonstrated in similar sensing devices in our research group [7]. In particular, the absence of solvent in the strip test leads to aggregation phenomena for the fluorescent probe on the solid surface, leading to a different emission response to the presence of the analyte with respect to the classic solution measurements.

Selectivity is a critical parameter for a real sensor to avoid false positive responses. To validate the selectivity of this strip test toward DA and NE, we measured its response to common interferents in saliva, such as glucose (2 nM), uric acid (200 μM), creatinine (20 μM), NaNO_2_ (200 μM), urea (3.33 mM), KNO_3_ (200 μM), and KH_2_PO_4_ (2.42 mM), usually used to mimic the ingredients of real saliva [31]. Figure 6 shows the possibility of efficiently recognizing NE with respect to the other interferents.

We tested the possibility of restoring the strip test by washing the device with water, where DA and NE are soluble (Figure 6). In particular, Figure 7 highlights the high number of cycles (>5) that can be carried out with the same device without any loss of efficiency, both for DA and NE sensing.

## 3. Materials and Methods

### 3.1. General Experimental Methods

The NMR experiments were carried out at 27 °C on a Varian UNITY Inova 500 MHz spectrometer ((International Equipment Trading Ltd., Mundelein, IL, USA) (^1^H at 499.88 MHz and ^13^C NMR at 125.7 MHz) equipped with a pulse field gradient module (Z axis) and a tunable 5 mm Varian inverse detection probe (ID-PFG). The ESI mass spectra were acquired on API 2000 (ABSciex, Framingham, MA, USA) equipment using CH_3_CN (in positive ion mode). Luminescence measurements were taken using a Cary Eclipse fluorescence spectrophotometer with a resolution of 0.5 nm at room temperature. The emissions were recorded at 90° with respect to the exciting line beam using 5:5 slit widths for all measurements. All the chemicals were reagent-grade and were used without further purification.

Hyperspectral Ultraviolet-Induced–Visible Fluorescence Mapping (HUVFM) was conducted using a custom-built instrument. The analysis probe consisted of a bundle of 19 Y-shaped fibers (BF19Y2HS02 sourced from Thorlabs, Ely, UK), provided with a beam collimator (F220SMA-532 sourced from Thorlabs) and separated by a watch glass from the analysis point. Among the 19 fibers, 10 were Y-ends connected to the source, which was a 365 nm LED sourced from Thorlabs. The remaining 9 fibers were linked to an optical block housing a bandpass filter designed to block backscattered light from the source. Subsequently, the CCD detector (CCS100/M, sourced from Thorlabs) was connected to a bundle of optical fibers arranged linearly (BFL200HS02 sourced from Thorlabs), maximizing the light collected by the sampling led.

### 3.2. Synthesis of (***2***)

BDPy-CH_2_Cl (**1**) [32] (150 mg, 0.42 mmol) was solubilized in 60 mL of dry acetonitrile; then, 1 eq. of KI (70 mg, 0.42 mmol), 4 eq. of K_2_CO_3_ (232 mg, 1.68 mmol), and 2 eq. of 2-picolylammine (0.08 mL, 0.84 mmol) were added at room temperature under a nitrogen atmosphere. The reaction mixture was refluxed overnight until the complete conversion of the BDPy-CH_2_Cl, monitored by TLC (silica gel, diethyl ether/AcOEt 6:4). The mixture was cooled at room temperature, and the solvent was removed under reduced pressure. The residue was redissolved in 50 mL of CH_2_Cl_2_ and filtered to remove K_2_CO_3_; then, the product (yield 40%) was isolated by column chromatography (silica gel, diethyl ether/AcOEt 6:4). ^1^H NMR (500 MHz; CDCl_3_): δ 1.03 (t, *J* = 7.9 Hz, 6H), 2.38 (s, 6H), 2.41 (q, *J* = 7.9 Hz, 4H), 2.50 (s, 6H), 4.01 (m, 4H), 7.18 (t, *J* = 6.4 Hz, 1H), 7.34 (d, *J* = 8.2 Hz, 1H), 7.65 (t, *J* = 8.2 Hz, 1H), 8.55 (d, *J* = 6.4, 1H) ppm. ^13^C NMR (125 MHz; CDCl_3_): δ 158.9, 153.4, 149.3, 138.3, 136.5, 131.8, 122.6, 55.6, 44.9, 17.1, 14.7, 12.4 ppm. ESI-MS: *m*/*z* = 422.1 [M − H]^−^. Anal. Calcd. for C24H31BF2N4: C, 67.93; H, 7.36; N, 13.20. Found: C, 67.87; H, 7.31; N, 13.12.

### 3.3. The Procedure for the Fluorescence Titrations

Two mother solutions of the receptor and catecholamine (1 mM) in dry solvent (CHCl_3_) were prepared. From these, different solutions with different ratios of receptor/guest were prepared. Fluorescence titrations of (**2**) and DA and NE were carried out using λ_ex_ = 500 nm, recording at λ_em_ = 550 nm and at 25 °C. With these data treatment, the apparent binding affinities of the receptors with DA and NE were estimated using HypSpec (version 1.1.33) [33,34,35]. The effect of the solvent was evaluated for each titration, excluding the interaction of chloroform with the BDPy probe.

### 3.4. Stoichiometry

Stoichiometry of the complexes was conducted using Job’s plot method and using spectrophotometric measurements. The samples were prepared by mixing equimolecular stock solutions (1 mM) of (**2**) and DA or NE to cover the whole range of molar fractions, keeping the total concentration (1 × 10^−6^ M) constant. The changes in absorbance compared to the uncomplexed receptor species (ΔA × χ^−1^) were calculated and reported versus the receptor mole fraction (χ). These plots invariably show the maximum at a 0.5 mol fraction of the receptor, thus suggesting its 1:1 complex formation.

### 3.5. Procedure for the Strip Test

Three solutions at different concentrations (from 10 μM to 1 nM) of DA and NE were prepared in MilliQ water. Three strip tests in polyamide (previously activated by exposing it to an oxidating environment produced by UV/O_3_-based surface treatment) were exposed to each solution, and the emission spectra before and after the exposure were acquired, as reported in the Appendix A. A control was also registered, consisting of MilliQ water. For statistical treatment, the following formula was applied, (I_water_ − I_analyte_)/I_BDPy_, where I_analyte_ is the emission of probe (**2**) after exposure to the analyte (DA or NE), I_water_ is the emission of probe (**2**) after exposure to MilliQ water, and I_BDPy_ is the emission of each probe before exposure to any analyte. All measures are the result of an average of three replications.

### 3.6. Recovery

The recovery of the strip sensor was tested by performing an analyte (DA and NE)-washing cycle with water (pH = 7). First, DA and NE (1 μM in MilliQ water) were deposited using a polycellulose cartridge onto probe (**2**), which was air-dried for 30 s, and the emission spectrum was acquired with the optical fiber. Subsequently, MilliQ water was deposited using the polycellulose cartridge onto probe (**2**) so that the analyte (water-soluble) and not the probe, which is not water-soluble, was solubilized, and the emission spectrum was acquired with the optical fiber. This cycle was repeated five times.

### 3.7. Computational Details

Ab initio and density functional calculations were performed using the Gaussian09 program package [36]. Optimization of all the involved systems was performed at the B3LYP/6-31G(d,p) level of theory. All the structures were subjected to a full conformational search to ensure reporting the absolute minimum and, if applicable, taking into account the Boltzmann distribution for conformers with ΔE < 1 kcal/mol. Frequencies were calculated and checked to make sure that all of them were positive and no imaginary frequencies were present. The Mulliken partial charges were calculated for each system at the B3LYP/6-31G(d,p) level of theory. GaussView 5.0.8 software was used as the graphic interface to visualize the proposed geometries. Zero-point energy (ZPE) was included in each result.

## 4. Conclusions

A new BODIPY (**2**) fluorescent sensor, bearing a picolylamine arm, able to recognize dopamine and norepinephrine by non-covalent interactions, has been reported. Recognition studies were performed using fluorescence titrations, showing good affinities for dopamine and adrenaline. The formation of the supramolecular complexes was confirmed by the DFT calculations, which suggested the formation of multiple hydrogen bonds. The strip test confirmed the possibility of obtaining a reusable solid device for the detection of norepinephrine, even in complex matrixes, such as human saliva. This study represents a proof of concept for the realization of real point-of-care devices for practical catecholamine detection.

## Data Availability

Compounds (**1**) and (**2**), as well as the data used in this work, can be obtained from the authors upon formal request.

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
