# Peer review of "A New BODIPY-Based Receptor for the Fluorescent Sensing of Catecholamines"

_molecules, 2024, doi:10.3390/molecules29153714_

Round 1

Reviewer 1 Report

Comments and Suggestions for Authors

The paper by Puglisi et al. describes a new BODIPY-based fluorescence sensor for dopamine and norepinephrine. Synthesis of the BODIPY-picolylamine conjugate is straightforward and the observed fluorescence intensity changes upon addition of  dopamine or norepinephrine would be of interest in view of possible sensor applications. I recommend publication after making the following revisions. 

1) Page 2, line 25: "To the best of our knowledge, no example of molecular sensors able to detect DA and NE simultaneously are reported" should modified because such a sensor is already known in literature (L. Zhang et al. Angew. Chem. Int. Ed. 2019, 58, 7611).  

2) I recommend that the authors show changes in the UV-vis absorption spectra upon addition of dopamine and norepinephrine. Spectral changes would be observed if the host-guest complexes (DA@2 and NE@2) are formed in solution. 

3) The fluorescence intensity is increased upon addition of dopamine in CHCl3 (Figure 2a), while the intensity is decreased in Strip Test (Figure 7). What caused the difference? The authors should add comments on this point. 

4) Page 8, line 4 from the bottom: The 1H NMR chemical shifts in CDCl3 are reported in the manuscript, while Figure S1 shows the 1H NMR spectrum in CD3OD. Please unify the 1H NMR data.

5) Figure S3: The peak position of the mass spectra (m/z = 422.1) is inconsistent with the data reported in the manuscript (m/z = 424.2). 

Author Response

The paper by Puglisi et al. describes a new BODIPY-based fluorescence sensor for dopamine and norepinephrine. Synthesis of the BODIPY-picolylamine conjugate is straightforward and the observed fluorescence intensity changes upon addition of  dopamine or norepinephrine would be of interest in view of possible sensor applications. I recommend publication after making the following revisions. 

Point 1: Page 2, line 25: "To the best of our knowledge, no example of molecular sensors able to detect DA and NE simultaneously are reported" should modified because such a sensor is already known in literature (L. Zhang et al. Angew. Chem. Int. Ed. 2019, 58, 7611).  

Response 1: we thank the reviewer for the comment. In accordance with this, the manuscript was revised by reporting the suggested reference (page 2, line 25-30 and ref. 25 in the new version).

Point 2: I recommend that the authors show changes in the UV-vis absorption spectra upon addition of dopamine and norepinephrine. Spectral changes would be observed if the host-guest complexes (DA@2 and NE@2) are formed in solution. 

Response 2: We agree with the reviewer and UV-Vis measurments for the (2)@DA and (2)@NAdr complexes were performed, but unfortunately no significant change in the absorption spectrum is apparent; this can be explained by the fact that the supramolecular complex probably occurs on the excited state and thus we are only able to see a change in emission following increasing additions of DA and Nadr. Main text has been modified according to these results (page 3, line 27).

Point 3: The fluorescence intensity is increased upon addition of dopamine in CHCl3 (Figure 2a), while the intensity is decreased in Strip Test (Figure 7). What caused the difference? The authors should add comments on this point. 

Response 3: we agree with the reviewer. A possible explanation has been provided in the new version of the manuscript (page 7, before Figure 5).

Point 4: Page 8, line 4 from the bottom: The 1H NMR chemical shifts in CDCl3 are reported in the manuscript, while Figure S1 shows the 1H NMR spectrum in CD3OD. Please unify the 1H NMR data.

Response 4: we thank the reviewer for the comment. According to this point, we have corrected the text.

Point 5: Figure S3: The peak position of the mass spectra (m/z = 422.1) is inconsistent with the data reported in the manuscript (m/z = 424.2). 

Response 5: We agree with the referee we modified the text because the mass spectrum was acquired in the negative ion mode, so what we observe is: m/z = 422.1 [M-H]-

Reviewer 2 Report

Comments and Suggestions for Authors

NMR data should be carefully check.

Author Response

The authors report a new fluorescent bodipy receptor bearing a picolylamine arm, which was able to interact by non-covalent interactions with DA and NE. Sensing studies were also performed in solution by fluorescence titrations, supported by DFT calculations. Strip test was fabricated to detect these catecholamines also on solid state, valuating the selectivity respect to other analytes contained in human saliva. In my opinion, this paper has the possibility of the publication in, so I recommend publication in Molecules while more discussions and revisions are required. The following aspects need to be clarified:

Point 1: On page 3, the photophysical properties of BODIPY compounds 1 and 2 should be summarized and discussed in detail within a table. The table should include properties such as absorption (Abs), emission (Em), quantum yield (Qy), Stokes shifts, and lifetime.

Response 1: according to the reviewer, a table summarising the photophysical properties of BDPy (1) and (2) has been added in the new version (Table 1, page 3).

Point 2: On page 3, you mention that Figure 1 shows the absorption and emission spectra of a 1 mM solution. However, in the caption of Figure 1, you indicate a concentration of 1 μM in CHCl₃. Could you please clarify if there is a typo?

Response 2: We thank the referee for the comment. We have corrected the text (page 3, line 9).

Point 3: With regard to the NMR data, I disagree with the analysis and the descriptions of the HNMR and CNMR spectra in SI, which do not match. BNMR and FNMR should be included to confirm the BF2 unit. Additionally, the spectra lack peak picking and integration. Furthermore, when presenting the carbon spectrum data, please round to one decimal place.

Response 3: we thank the reviewer for the comment. 1H NMR characterization of compound (2) has been revised (some integral values relative to the aromatic region were uncorrected). The presence of BF2 unit has been confirmed by ESI-MS measurement (see the Supplementary materials): in fact, the presence of a peak at m/z = 422 is undoubtedly relative to the molecule including BF2 unit. Peak picking and integral values have been reported in the revised Supplementary Materials. In addition, also 13C NMR characterization of (2) has been modified according to the reviewer’s comment.

Reviewer 3 Report

Comments and Suggestions for Authors

In this manuscript, the author reported the “New Bodipy-based receptor for the fluorescent sensing of catecholamines”.  The experiments have been well performed. ã€€The manuscript can be publishable in molecules after the following minor points are fully revised.

1) The 1H NMR integral value for sensor 2 is incorrect.

1H NMR (500 MHz; CDCl3): d 1.03 (t, J = 7.9 Hz, 6H), 2.38 (s, 6H), 2.41 (q, J = 7.9 Hz, 4H), 2.50 (s, 6H), 4.01 (d, J = 3.6 Hz, 2H), 7.18 (t, J = 6.4 Hz, 2H), 7.34 (d, J = 8.2 Hz, 2H), 7.65 (t, J = 8.2 Hz, 2H), 8.55 (d, J = 6.4, 2H) ppm.

2) Show the integral value in the1H NMR spectrum shown in Figure S1.

Author Response

In this manuscript, the author reported the “New Bodipy-based receptor for the fluorescent sensing of catecholamines”.  The experiments have been well performed. The manuscript can be publishable in molecules after the following minor points are fully revised.

Point 1: The 1H NMR integral value for sensor 2 is incorrect.

1H NMR (500 MHz; CDCl3): d 1.03 (t, J = 7.9 Hz, 6H), 2.38 (s, 6H), 2.41 (q, J = 7.9 Hz, 4H), 2.50 (s, 6H), 4.01 (d, J = 3.6 Hz, 2H), 7.18 (t, J = 6.4 Hz, 2H), 7.34 (d, J = 8.2 Hz, 2H), 7.65 (t, J = 8.2 Hz, 2H), 8.55 (d, J = 6.4, 2H) ppm.

Response 1: we thank the reviewer for the comment. 1H NMR characterization of compound (2) has been revised (some integral values relative to the aromatic region were uncorrected).

Point 2: Show the integral value in the1H NMR spectrum shown in Figure S1.

Response 2: according to the comment, integral values in the revised Supplementary Materials have been reported.

Round 2

Reviewer 2 Report

Comments and Suggestions for Authors

agree to publish